# Role of B Cells in Mycobacterium Tuberculosis Infection

**DOI:** 10.3390/vaccines11050955

**Published:** 2023-05-06

**Authors:** Paul Stewart, Shivani Patel, Andrew Comer, Shafi Muneer, Uzma Nawaz, Violet Quann, Mira Bansal, Vishwanath Venketaraman

**Affiliations:** Department of Basic Sciences, College of Osteopathic Medicine of the Pacific, Western University of Health Sciences, Pomona, CA 91766, USA; paul.stewart@westernu.edu (P.S.); shivani.patel@westernu.edu (S.P.);

**Keywords:** humoral immunity, tuberculosis, mycobacterium tuberculosis, B cell, vaccines

## Abstract

Historically, research on the immunologic response to *Mycobacterium tuberculosis* (*M. tb*) infection has focused on T cells and macrophages, as their role in granuloma formation has been robustly characterized. In contrast, the role of B cells in the pathophysiology of *M. tb* infection has been relatively overlooked. While T cells are well-known as an essential for granuloma formation and maintenance, B cells play a less understood role in the host response. Over the past decade, scarce research on the topic has attempted to elucidate the varying roles of B cells during mycobacterial infection, which appears to be primarily time dependent. From acute to chronic infection, the role of B cells changes with time as evidenced by cytokine release, immunological regulation, and histological morphology of tuberculous granulomas. The goal of this review is to carefully analyze the role of humoral immunity in *M. tb* infection to find the discriminatory nature of humoral immunity in tuberculosis (TB). We argue that there is a need for more research on the B-cell response against TB, as a better understanding of the role of B cells in defense against TB could lead to effective vaccines and therapies. By focusing on the B-cell response, we can develop new strategies to enhance immunity against TB and reduce the burden of disease.

## 1. Introduction

TB is caused by an airborne pathogen of the organism *M. tb* [1]. *M. tb* primarily affects the lungs but can cause disease in any part of the body [1]. The lung is the portal of entry for most cases of TB with studies suggesting pulmonary involvement in the order of 70 to 90% of infections [2]. Infection of *M. tb* can lead to the bacteria being isolated within a granuloma, also known as latent TB [1]. Latent TB (LTB) infections are seen in individuals infected with *M. tb* but experiencing no clinical symptoms of active TB (ATB) [3]. Only ATB is contagious [1]. In 2018, the World Health Organization estimated that 1.5 million people died from TB, a higher mortality rate than any other single infectious agent [4]. It is estimated that 5–30% of the TB burden is from recurrent pulmonary TB [5]. Recurrent pulmonary TB is a proxy of TB drug resistance and community control of infection [5,6].

Despite the significant burden of disease, the only effective TB vaccine available is bacillus Calmette-Guérin (BCG) [7]. The BCG vaccine has been used since 1921 [8]. Results show that the protective effect of the BCG vaccine against the pulmonary form of TB is inconsistent with 0–80% efficacy [8]. The lack of efficient protection against TB suggests that Th1 cytokines and the innate immune response are helpful but not exhaustive for immunity against TB [9]. There is relatively little research on the role of B cells and the production of antibodies (Abs) in the humoral defense against TB compared to the cell-mediated response of macrophages and T cells. This is due to the widely held belief that TB is primarily a cell-mediated immune response, with T cells and macrophages playing a pivotal role in recognizing and attacking the bacteria [10,11]. It is commonly believed that *M. tb* is an intracellular pathogen, meaning that it can survive and replicate inside host cells. Therefore, humoral immunity provided by B-cell-mediated Abs production is thought to be inefficient in accessing and harming intracellular bacteria [12]. However, recent studies have suggested that B cells may play a more significant role in defense against TB than previously thought.

## 2. Pathophysiology in TB Infection

### 2.1. Innate Immune Response

Anatomical and physiological barriers provide the first line of defense against invading pathogens. The subsequent line of protection is the innate immune response, which includes macrophages, neutrophils, mast cells, natural killer cells, dendritic cells (DC), and eosinophils [10,11]. Once *M. tb* invades, the innate immune cells recognize *M. tb* by pathogen-associated molecular patterns (PAMP) via an immune cell pattern recognition receptor (PRR). Within minutes of *M. tb* exposure, the innate immune cells create an inflammatory response [10]. Bacteria, such as *M. tb,* are especially pathogenic due to their ability to evade host immune defenses. As an intracellular pathogen, *M. tb* must gain access to dividing immune cells, such as macrophages and DC, to divide [13].

After inhalation of *M. tb*, the pathogen reaches the alveoli (Figure 1A) [14], where macrophages, DC, and other innate immune cells recognize PAMP by toll-like receptors (TLR) found on most innate immune cells [15]. The TLR will recognize the PAMP of *M. tb* and activate proinflammatory cytokines, including IL-12 and nitric oxide [15,16,17]. They are then ingested by alveolar macrophages (Figure 1B) [14]. *M. tb* is then released into the surrounding lung tissue, which can lead to further infection of host immune cells. In addition, once released into the lung airspace, *M. tb* can become aerosolized and infect new hosts via air droplets containing the bacteria [13]. Lastly, *M. tb* contains TB necrotizing toxin (TNT), which leads to the loss of NAD+ in the *M. tb*-containing macrophages, which inhibits necrotic cell death of the macrophage (Figure 1C) [18].

### 2.2. Cell Mediated T-Cell Immune Response

The adaptive immune response is an antigen-specific immune surveillance system that recognizes and responds to pathogenic antigens and is regulated by crosstalk among innate immune cells [19]. While the innate immune system has developed to eliminate pathogens rapidly, the adaptive immune response was developed to overcome a foreign pathogen’s genetic variations and mutation ability [20]. The adaptive immune response primary functions are recognition of self-antigens from pathogenic antigens, elimination of pathogens/pathogenic cells, and development of immunologic memory [11]. The adaptive immune cells are as follows: (1) B cells produce Abs. (2) T cells are activated by B cells and antigen-presenting cells (APC) [11].

Once *M. tb* antigens are phagocytized, they are presented to DC to initiate an adaptive immune response [21]. The adaptive immune response begins the moment DC presents *M. tb* inside the lymph nodes, where CD4+ T cells will be activated and migrate to the lungs to help stop the growth of *M. tb* [15,21]. In vivo, human models of patients with HIV CD4+-depleted T cells show striking evidence of the importance of these cells in TB immunity [22]. Furthermore, patients with ATB had routinely decreased proportions of polyfunctional CD4 T cells producing IL-2 and interferon-gamma (IFN-γ) when compared to individuals with LTB [23,24]

Once a granuloma is formed, the immune response can reduce bacterial multiplication leading to LTB infection (Figure 1D) [14]. Moreover, tumor necrosis factor-alpha (TNF-α), a cytokine produced by T cells, plays a crucial role in granuloma formation and has immunoregulatory properties [15]. Observational studies have shown that humans treated with TNF-α inhibitors are more susceptible to TB [25]. However, TNF-α may also contribute to unwanted inflammatory responses as the disease progresses [15].

Similar research for CD8 T cells is limited because there are few situations where humans deficient in CD8 T cells are in contact with *M. tb*. However, studies have shown a direct correlation between the amount of granzyme A released and the degree of growth restriction of *M. tb* [26]. Furthermore, the depletion of CD8 T cells in the chronic stage of infection in mice resulted in an increased bacterial burden, suggesting these cells are necessary for long-term infection control [27,28]. The mechanism of action may be due to CD8 T cells recognizing the major histocompatibility class I (MHC I) and producing IL-2, IFN-γ, and TNF-α, which have a prominent role in controlling *M. tb* [22]. However, data suggest higher levels of regulatory cytokines produced by CD8 T cells, such as IL-10 and TGF-β in patients with ATB, were associated with a higher bacterial burden [27,29].

### 2.3. Humoral B-Cell Immune Response

Humoral immunity is the function of B lymphocytes. B lymphocytes may be called to transform into plasma cells to produce Abs [30]. The role of Abs includes neutralizing infectious agents and antibody-dependent cellular cytotoxicity (ADCC). Furthermore, secreted Abs can serve to activate the complement system and further enhance bacterial phagocytosis and apoptosis of infected host cells [31]. Some B cells produce immunological memory cells, which are defined as long-lived cells that rapidly respond to previously encountered pathogens upon recall [32,33]. B lymphocytes are also known to promote recruitment and activation of other immune cells via antigen presentation and cytokine production.

Emerging evidence supports a role for B cells and the humoral response in pathogens whose life cycle requires an intracellular environment, such as chlamydia trachomatis, salmonella enterica, *Francisella tularensis*, etc. [12]. While the direct role of Abs has been unclear. A recent study in 2016 of individuals with LTB and ATB infections shows divergent humoral signatures [34]; moreover, the same study showed that individuals with a LTB infection had a unique Ab Fc functional profile and a distinct Ab glycosylation pattern [34]. Furthermore, when comparing Abs from ATB to LTB, Abs from LTB had enhanced phagolysosome maturation, inflammasome activation, and macrophage killing of intracellular *M. tb* [34]. Given these results, understanding the extent of humoral immunity in the disease process can lead to therapeutic targeting.

### 2.4. Germinal Centers

Germinal centers (GC) are the sites for Abs diversification, maturation, and antigen-dependent clonal expansion. Thus, they are essential to humoral immunity [35]. GC are known to be present in secondary lymphoid tissues, including lymph nodes, the tonsils, the spleen, Peyer’s patches, and mucosa-associated lymphoid tissue (MALT) [36]. B-cell selection from the GC has four potential outcomes: apoptosis, further processing, or differentiation into memory B cells or plasma cells [37]. Histological studies show that GC is divided into two anatomically distinct regions: the dark zone (DZ) and the light zone (LZ) [38]. The DZ contains mitotically active B cells known as centroblasts, where GC-B cells extensively proliferate [38,39]. The LZ has non-dividing B cells known as *centrocytes* [38]. The LZ-B cells will receive an antigen via a follicular dendritic cell (FDC). After binding to the antigen, the LZ- B cell will internalize the B-cell receptor antigen, creating a major histocompatibility class II (MHCII), which enables them to receive help from T follicular helper cells [39]. Early theories on GC processing focused on antigens and FDC binding for the selection of B cells [37]. However, discoveries have shown that CD4-positive T helper cells surround the area where FDC and antigen binding occurs. Therefore, T helper cells and the cell-mediated response are vital in regulating GC selection [37]. A study by Victoria et al. using a photoactivatable green fluorescent protein (PA-GFP) with flow cytometry showed that B cell movement from the LZ to the DZ is controlled by T cells based on the amount of antigen captured [38]. A follow-up from this study found that the amount of antigen during these interactions with T cells affects the downstream proliferation of B cells [37]. Overall, this shows the interplay and communication between the cell-mediated and humoral immune responses in developing GC-B-cell differentiation and proliferation.

While GC have been characterized extensively in secondary lymphoid tissue, their presence in TB granulomas is a relatively new finding [40]. GC are a distinctly B-cell-derived structure and their presence in TB granulomas suggests that B cells may play a role of questionable significance in the immunologic response to a TB infection [41]. While granulomas are a characteristic histologic finding in TB, GC is seldom discussed due to the primarily cell-mediated immune response to TB [42,43]. The recent findings of GC as a consistent histological feature in TB granulomas have led to comparing of these GC to those found elsewhere in the body [44].

## 3. Granulomas in TB

During the acute phase of TB, *M. tb* infects the host airway immune cells, evades phagocytosis, and persists and replicates intracellularly leading to host cell necrosis and intercellular dissemination of the pathogen. To prevent further dissemination of *M. tb*, the host immune system responds with granuloma formation, the hallmark of *M. tb* infection. Granulomas are defined as aggregates of immune cells that produce various cytokines to limit bacterial growth. Granuloma formation begins when infected macrophages produce cytokines leading to the activation of CD4+ T lymphocytes, which then produce proinflammatory cytokines, such as IFNγ and TNFα [45]. These cytokines, in turn, lead to the M1 differentiation of macrophages, a proinflammatory state that leads to the formation of granulomas [46]. Although this process is thought to be driven primarily by macrophages and T cells, new evidence suggests the humoral immune response plays a vital role in activating the immune response against *M. tb* and granuloma formation.

Phuah et al. (2012) used several biomarkers to characterize the composition of the granulomas found in nonhuman primates (NHP) infected with *M. tb* and demonstrated that lung granulomas of infected NHPs contained significantly more activated B cells compared to noninfected NHPs [47]. A study by Hunter et al. (2022) showed similar findings in which a higher number of B cells were seen in granulomas collected from cynomolgus macaques at four and twelve weeks post TB infection compared to granulomas from rhesus macaques [48]. Classically, cynomolgus macaques are known to better control disease progression compared to rhesus macaques. The results of the study show that rhesus macaques, with fewer granuloma-associated B cells, had a worse outcome and a higher disease burden at post 4 and 12 weeks of TB infection. To further understand the role of B lymphocytes within the granuloma, researchers sought to characterize the structure of immune cells within the granuloma.

Several studies suggest the granuloma structure formed during TB infection closely resembles that of GC found in secondary lymphoid tissues. Various studies have highlighted structural and functional similarities between the GC in secondary lymph nodes and granulomas in TB infection. Structurally, granulomas contain a necrotic core surrounded by a layer of immune cells that form a physical barrier to contain the infection. Phuah et al. used several biomarkers to study the structure of granulomas in NHP and found CD20+ B cells clustered toward the periphery of granulomas similar to that of GC [47]. The B cells within the granuloma were further stained for PNA expression, a marker for GC-specific B cells, anti-human Ki-67, a marker for B-cell proliferation, and PNAd, a marker for high endothelial venules found in GC, to characterize further the similarities between the B cells found in granulomas and GC. Although no PNA staining was found in the B-cell clusters of granulomas, PNA staining was positive in the macrophage-dense areas of granulomas [47]. However, PNA was found to be a nonspecific marker for GC-associated B cells in NHP models and, therefore, was excluded from the study as an accurate marker for GC. Instead, PNAd was used to compare the structures of TB granulomas in NHP and GC. PNAd positive cells were found to be near B-cell clusters within the granuloma, similar to B cells found in GC. To further highlight the structural similarities between GC and granulomas, Ki-67 positive cells were found within the B-cell clusters of the granulomas. However, unlike GC, only a few B cells within the granuloma were shown to be actively proliferating compared to the entire B-cell cluster proliferation within GC. Nonetheless, B-cell clusters within granulomas show distinct similarities to B-cell structures found in secondary GC (Table 1).

Given the structural similarities between B cells found in GC and TB granulomas, further studies were conducted to better understand the role of B cells within granulomas [57]. Choreño-Parra et al. (2020) found that the B cells in follicle-like structures within the granulomas of *M. tb-*infected mice served various complex functions in eliciting an immune response against TB. B cells within granulomas were found to produce the chemokine CXCL13, which recruits CXCR5+ T cells to the infection site, and promotes formation of follicle-like structures within granulomas [57]. To confirm the suggested role of B cells in the immune response against TB, B-cell-deficient mice were infected with aerosolized *M. tb* and found to have an increased bacterial burden, enhanced inflammation of lung tissue, and a decreased number of CXCR5+ T cells within granulomas. The B cells found in *M. tb*-infected lungs were also shown to upregulate the expression of Sftpc and Ccr7. Sftpc is an important gene that codes for a surfactant protein usually found in lung epithelial cells. B-cell expression of Sftpc confirms their localization to lung parenchyma during an ATB infection. Ccr7 is a vital chemokine that promotes B-cell follicular development within TB granulomas [57]. Similar to B cells found in GC, the B cells found in the follicular structures in TB granulomas demonstrate a similar function in serving as antigen presenting cells, promoting T-cell and plasma cell proliferation, and eliciting a protective immune response against *M. tb* [58].

## 4. Acute Infection

The acute phase of TB is primarily understood to be a T-cell- and macrophage-driven process [59]. However, recent studies have shown that B cells are actively involved in this phase of infection. B-cell depletion has been shown to retard the progression of many T-cell-mediated autoimmune diseases, such as multiple sclerosis and type I diabetes [45], which highlights the important role of B cells in activating T cells. Furthermore, a lack of the cellular immune response was found to be correlated with an increase in the humoral immune response in a *M. tb* infection [45]. Although protection against intracellular pathogens, such as *M. tb*, has been thought to exclusively elicit a cellular immune response, B cells and the humoral immune response help complement the cellular immune response to contain and overcome intracellular infections.

In addition to acting as antigen presenting cells, B cells function to produce Abs directed against pathogens. Once activated, *M. tb-*directed Abs can regulate effector functions, such as neutralizing antigens, opsonization, and ADCC [45]. Although *M. tb* is a non-toxin-producing, intracellular pathogen, neutralizing Abs are thought to bind to epitopes on *M. tb* and play an essential role in the Abs-mediated clearance of *M. tb* [60]. Abs produced by B cells are also able to bind antigens to pathogens and tag them for phagocytosis [61]. A recent study showed that immunoglobulin G (IgG)-dominated opsonization of *M. tb* led to increased phagocytosis by macrophages and enhanced complement binding of C3 and C4 [62]. Although *M. tb* is primarily an intracellular pathogen, *M. tb* re-enters the extracellular space when infecting other host cells. During this phase, IgG ADCC limits growth of *M. tb* and plays an important role in host defense [63]. Purified protein derivative (PPD)-specific IgG Abs were shown to increase natural killer cell-mediated ADCC in patients with both ATB and LTB [34]. Abs have been shown to positively augment the immune response against *M. tb* through various mechanisms while supporting the cellular immune response [64].

In the clinical setting, passive administration of human IgG has been shown to control *M. tb* growth and inflammation of the lungs [62,65]. Phuah et al. showed that plasma cells produced *M. tb*-specific Abs within granulomas. Furthermore, infected tissues contained higher levels of TB-specific IgG. These findings show that B cells are not only present but actively generating *M. tb*-specific Abs at the site of infection [45,47]. In addition, the anti-inflammatory activity of IgG has been proven and used in intravenous immune globulin (IVIG) in pathologies, such as bacterial sepsis and HIV infection [45]. Research by Lyashchenkoa found similar results on the importance of B cells in antigen presentation and added that B cells also produce IL-12 and IFN-γ that activate macrophages and DC, enhancing their ability to kill intracellular microbes [66]. Despite these results, the use of Abs as a means of protection is conflicting. A study in 2015 showed that B cells had no significant contribution in protective immunity in TB [67]. In addition, there is no research linking the use of CD20+ inhibitors and an increased risk of TB infection [67]. Currently, IgG-specific directed Abs against specific *M. tb* antigens have no role in clinical therapy or prophylaxis and a minor role in TB vaccines [67]. As such, assessing the use of Abs in observational studies and experiments in acute and chronic stages of ATB would be of great value, thus, creating better tools and therapies for TB management.

Studies have shown that IL-6 inhibition leads to an increased risk of TB; it is often considered one of the essential cytokines during infection [68,69]. IL-6 is involved in proinflammatory and anti-inflammatory responses and its functions include the differentiation of monocytes into macrophages by controlling the expression of macrophage colony-stimulating factor, promoting Th-2 response by inhibiting Th-1, increasing B-cell Ig-G production, and inducing CD4 into Th-17 [69]. Various cells, including T cells, B cells, and macrophages, produce IL-6. Therefore, a study by Irina Linge et al. (2022) assessed the impact of IL-6 produced by B cells in the lungs of infected *M. tb* mice. To investigate the impact of IL-6 produced by B cells, a mouse strain with specific IL-6 deficiency in B cells was established on the B6 genetic background [49]. Mice with ablation of IL-6 in B cells (B-IL-6KO) were generated by crossing IL-6^flox/flox^ mice [23] with CD19-Cre knock-in mice (CD19^cre/cre^) [49]. The study revealed that B-IL-6KO had a 10-fold decrease in total IL-6 mRNA in the lungs compared to wild-type mice (Figure 2A) (Table 1). This decrease in IL-6 led to an increased susceptibility to TB in terms of the post-infection life span [49]. This suggests that B cells play a pivotal role in IL-6 production in a Tb infection. Moreover, this same study showed that B-IL-6KO mice had a significantly lower proportion of IFN-γ-positive lung CD4+ cells than the controls. However, these differences disappeared by the week seven post-challenge [49]. Finally, IL-6 from B cells was essential for Th17 differentiation and is required for CXCR5+ Tfh cells’ differentiation and maintenance (Figure 2A). Overall, Linge et al. showed that IL-6 of the B-cell origin is critical in the early phase of infection in the development of acquired anti-TB immunity in the lungs and explained how B cells and T cells interact in the early stage of an anti-TB response [49]. However, this study failed to show the value of IL-6 from B cells after the acute phase of the infection [49].

## 5. Rituximab

Rituximab is used clinically to treat systemic lupus erythematosus and multiple sclerosis [51,52]. Studies have shown that rituximab effectively depletes peripheral B cells by targeting CD20+ cells [53,54]. Although limited, some studies have shown that rituximab does not affect the plasma-containing compartment of B cells due to the little CD 20+ expressed in these cells [55]. Phuah et al. studied the effects of B-cell depletion via rituximab-treated macaques infected with *M. tb*. Then, they investigated B-cell depletion and its impact on granuloma T-cell cytokine secretions, such as IL-17, TNF-α, IFN-γ, IL-10, and other immunoregulatory cytokines [55]. Results showed that rituximab-treated animals had decreased IL-6 and IL-10 in their granulomas compared to the controls [55]. However, changes in levels of IL-17 and TNF-α that were found were not statistically significant (Table 1) [55]. The study concluded that rituximab-treated cells had little effect on the pathology of *M. tb*-infected macaques. By contrast, B cells significantly affected cytokines and inflammation levels [55]. Lastly, there were five times as many bacteria in the granulomas of those treated with rituximab compared to those without the treatment [55]. This data provided evidence of and insight into the role of B cells during *M. tb* infection. However, the study is limited regarding inclusion criteria as all patients were in the early phase of the *M. Tb* disease. Finally, a study by Cantini et al. aimed to evaluate TB reactivation in patients who have rheumatoid arthritis receiving rituximab and other non-anti-tumor necrosis factor agents [56]. Results showed that the TB risk was negligible for patients taking rituximab (Table 1) [56]. These conflicting results call into question the long term role of B cells and if B-cell KO mouse models are appropriate due to the altered structure of immune organs [50].

## 6. Chronic Infection

The role B cells play in the acute phase response to a TB infection has been corroborated by multiple sources; however, there is conflicting research on their role in the later phase of the disease. Whether their role in later infection is limited or even detrimental has yet to be solidified by research. An experiment by Suraj Parihar showed that IL-4-responsive B cells increased mice’s mycobacterial burden and lungs [70]. Most other research on the long-term role of B cells is speculative. As Simone A. Joosten suggests, B cells must play some role due to the decreased B cells in patients with ATB and recent LTB infection [44]. Those B cells that remain are phenotypically atypical and are termed exhausted B cells [44]. B-cell exhaustion is defined as the loss of the normal function of B cells as a consequence of chronic disease states [44]. Exhausted B cells are also named atypical B cells because they lack CD27 and CXCR5 expression on their surface [44]. Atypical B cells have an impaired ability to proliferate and produce cytokines or chemokines [44]. Because of this, atypical B cells are thought to be responsible for disease progression from latent to active disease [66]. This is further evidenced by the B-cell population returning to normal function and numbers after a course of Abs to treat the TB infection [44].

Linge et al. (2023) explored the role of B cells during late TB infection, after 16 weeks. Results showed that TB susceptible mice had a significant decrease in B cells and B-cell follicular formation at week 16 post infection, which led to an increase in inflammatory cytokines genes encoding IL-1, IL-11, IL-17, and TNF-α when compared to TB-resistant mice [50]. Moreover, chronic infection of TB in B-cell-depleted mice led to shorter life spans and increased mRNA expression of neutrophil recruiting factors CXCL1 and IL-1 and genes associated with neutrophil inflammation, such as matrix metalloproteinases MMP8 and MMP9, which ultimately led to severe lung pathology (Figure 2B) [50]. In addition, although IL-6 was shown to be decreased in the early stage of infection in B-cell-depleted mice, in the later stage of the disease, B-cell depletion led to an increase in IL-6 [50]. Linge et al. revealed that in the advanced stage of infection without B cells, macrophages are a significant source of IL-6 production [50]. Moreover, B-cell depletion in advanced TB resulted in an increased neutrophil response in the lung tissue of infected mice; this increased lung fibrosis compared to B-cell-rich mice [50]. The study showed that B-cell depletion in late-stage infection had little influence on the *M. tb* burden but enhanced cachexia and shortened the animals’ life span compared to controls (Figure 2C) (Table 1) [50]. Furthermore, the study revealed that B cells may be more protective in late-stage infection. Taken together, we can conclude that B cells provide a vital role in IL-6 production during the early phase of the infection and provide a protective role in the chronic stage of infection by attenuating proinflammatory cytokines in the lungs reducing fibrosis and other severe pathology.

Finally, a study of the histological structure of TB granulomas by Ulrichs et al. revealed a distinct importance of antigen-presenting B cells in a follicle-like formation in the granuloma’s periphery. This study concludes these peripheral B-cell follicles serve as a as morphological substrate, allowing for a sustained immune response by the host [71]. In chronic infection, these follicles may serve as a source of antigen presentation and Abs production [71].

## 7. Vaccines and B-Cell Response

Vaccines have played a crucial role in maintaining protection against some of the deadliest organisms in the world, including *M. tb*. Vaccine efficacy has been documented since the early 18th century with substantial advances. With new advances in research, B cells have been found to play a significant role in the physiological benefits of vaccines. Upon administration of a vaccine, antigen-presenting cells will introduce the novel antigen/protein to the body’s immune system. Subsequently, effector B cells will allow for the development and secretion of Abs against the specific antigen to prevent future infection and disease [72]. These Abs have been confirmed via Abs serum titer labs to remain within the host’s immune system for an extended period of time, further confirming that B cells are crucial to the evolution of a host’s humoral immunity [73].

Although not primarily utilized in the United States, the bacillus Calmette-Guerin (BCG) vaccine has been administered in other parts of the world to develop protection against *M. tb* and its disease processes. The efficacy of the BCG is controversial due to differences in strains among countries. However, it has still been proven to confer immunity to many, thereby preventing numerous deaths that may have otherwise occurred [74].

While a targeted antibiotic regimen has been developed and used in many of those afflicted with TB, clinicians and researchers still maintain that a more cost-efficient and effective way to combat *M. tb* would be the production of anti-TB vaccines [12]. This belief is further supported with conclusive evidence that both B cells and Abs participate in immunity against *M. tb*. Studies show that the passive transfer of monoclonal Abs against *M. tb* antigens can aim to enhance a host’s immune response should it be exposed to the pathogen [75]. In addition, various independent studies utilized three general criteria to cement the role of B cells in immunity against *M. tb*: (1) Passive transfer of monoclonal Abs against the *M. tb* antigen alters the disease course such that there is an overall benefit to the host; (2) the existence of those specific Abs leads to decreased susceptibility to the particular disease (TB); and (3) there is increased susceptibility to the particular disease (TB) in those who are deficient in the Abs against *M. tb*. [75].

The goal of a vaccine is to induce long-term immunological memory [76]. Despite the popular belief that vaccines protect against pathogens by eliminating foreign invaders, many successful vaccines protect against disease by giving the immune system a “head start” through immunological memory [76]. Memory B cells are crucial in vaccine development due to their long life span and rapid response to subsequent exposures to the same pathogen [77]. Furthermore, vaccine-induced antibodies can be maintained in the serum for extended periods by long-lived plasma cells residing in the bone marrow and constitutively producing antibodies without antigen [77]. Although current vaccines against TB elicit cellular immunity, more recent research suggests vaccines can be used to induce Abs-mediated immunity against Tb [63]. Aguilo et al. (2020) studied the effects of a novel inactivated vaccine, *M. tb* VAC HK, in mice and NHP. They found that the vaccine was able to recruit mucosal Abs and aid with *M. tb* opsonization and improve protection against TB [78]. Additionally, Prados-Rosales et al. (2017) created two polysaccharide-conjugate vaccines by linking arabinomannan (AM), a capsular polysaccharide found on the surface of mycobacteria, to *M. tb* Ag85b or B. anthracis protective antigen [79]. Their results demonstrated that immunization with either vaccine led to the production of Abs against *M. tb*-specific AM. Mice immunized with the vaccines were found to have a lower bacterial burden and a longer life span. Dijkman et al. (2019) further studied the role of the humoral immune response against *M. tb* specifically in the pulmonary mucosa of NHP [80]. Pulmonary mucosal delivery of the BCG vaccine was shown to reduce the effects of a TB infection compared to the standard intradermal delivery of the BCG vaccine. Pulmonary administration of BCG was shown to induce T helper type 17 (Th17) cells, interleukin-10 (IL-10), and immunoglobulin A (IgA) to promote local immunity against *M. tb* [80]. With these studies and their conclusions in mind, there is insurmountable evidence supporting the argument for more research on B cells and the humoral immune response to the *M. tb* disease process. The concluded interpretations of these independent studies therefore point to the postulation that anti-TB vaccines that induce Abs-mediated immunity can result in enhanced protection against the life-threatening disease.

## 8. Conclusions

This review aimed to determine the significance of the humoral immune response against *M. tb* infection. Despite the potential role of B cells in combating *M. tb*, studies on B cells and *M. tb* infection are limited. In contrast to the previous understanding of B cells and the humoral immune response against intracellular pathogens, we show that emerging research supports that B cells may attenuate the life span of many intracellular organisms, such as chlamydia trachomatis and salmonella enterica. Regarding TB, recent data have demonstrated divergent humoral signatures, unique Ab Fc function, and distinct Abs glycosylation patterns when comparing individuals with ATB and LTB infection. Furthermore, it has been shown that animals infected with TB produce *M. tb*-specific Abs in the lungs.

Moreover, new evidence suggests that the humoral immune response may partially drive granuloma formation, formerly thought to be exclusively a macrophages and T-cells-mediated process. Experiments in NHP demonstrated that granulomas with fewer B cells had worse outcomes and a higher disease burden. Furthermore, studies reveal striking similarities when comparing the structure of granulomas and GC, such as B cells found in the periphery of granulomas. However, unlike GC, these B cells were not shown to be actively dividing. In addition, B cells within granulomas produced CXCL13, which recruits CXCR5+ T cells to create follicle-like structures within the granuloma. We can conclude that granuloma formation and maintenance may require B cells. However, additional analysis and testing are needed.

The number of experimental models showing the importance of T cells and macrophages is exhaustive. Our review shows that B cells are actively involved in this phase of the disease by creating IgG-dominated opsonization leading to increased phagocytosis. In cytokine production experiments, B-IL-6KO strains had a significant decrease in total IL-6 production, leading to increased TB susceptibility. Finally, B-cell depletion in the chronic stage of infection resulted in neutrophil inflammation and contributed to significant lung fibrosis. Overall, this suggests that B cells help fight TB in the acute phase of infection by creating proinflammatory cytokines and Abs. However, in the later stage of the disease, B cells have an anti-inflammatory effect by decreasing lung fibrosis and neutrophil inflammation.

Despite these findings, several studies resulted in somewhat controversial conclusions, such as the rituximab study, which showed that TB risk was negligible for patients taking rituximab. Moreover, vaccine studies are even more conflicting, with some studies concluding that mouse monoclonal Abs showed no protection against a *M. tb* challenge. By contrast, more recent studies show that inactivated *M. tb* vaccines in mice and NHP were able to recruit antibodies that aid in the protection against TB. These conflicting results question the validity of mice and other animal studies of the B-cell response in a TB infection due to the divergent structure of immune organs compared to humans. Despite these controversial findings, a growing body of the literature shows the importance of B cells in combating a *M. tb* infection. Further research is needed to address these issues.

The role of B cells in TB is poorly understood, and research is often contradictory. Current research seems to suggest a phase-dependent role of B cells in fighting a TB infection. Most research is in agreement that B cells are critical for the initial phase of infection, including cytokine production that activates neutrophils and macrophages and the subsequent granuloma formation. Research is less detailed about how B cells behave after the first 13 weeks of infection. In conclusion, developing research suggests that B cells play a critical role in killing intracellular organisms by creating cytokines and Abs to help eliminate *M. tb* and protect the host. However, more research is vital to better understand the contribution of B cells to the immune response against TB. Therefore, the role of B cells in TB immunity may lead to the development of novel therapies to help combat the global TB threat.

## Figures and Tables

**Figure 1 vaccines-11-00955-f001:**
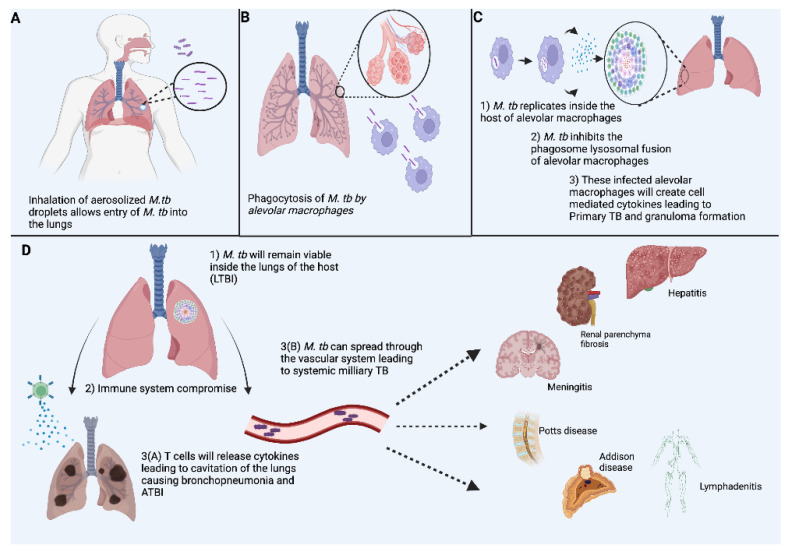
Overview of the manifestation and Pathophysiology in TB Infection. As outlined in (**A**–**D**), a *M. tb* infection travels through airway structures, eliciting innate immune defenses, including alveolar macrophages. *M. tb* hijacks key functionalities of the macrophages to aid in establishing a primary infection. Once established in its granulomas within the lungs, *M. tb* travels through vasculature to cause widespread disease throughout the body.

**Figure 2 vaccines-11-00955-f002:**
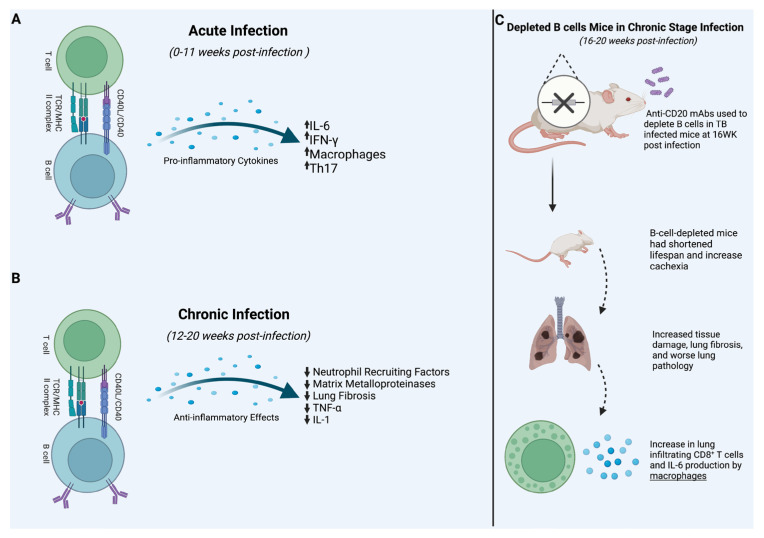
Effects of B-cell depletion in the acute and chronic stage of TB infection. As outlined in (**A**–**C**), acute and chronic stages of post *M. tb* infection elicit varying responses from the body’s immune system. In acute infection, B cells trigger first line defenses, including inflammatory cytokines, macrophages, and Th17. In chronic infection, B cells work to oppose inflammation and preserve healthy lung tissue through turning off proinflammatory signals and stimulating anti-inflammatory processes. When B cells have been depleted, lung tissue damage worsens resulting in the recruitment of cytotoxic CD8+ T-cells and the reproduction of IL-6 by macrophages leading to severe tissue damage and lung fibrosis.

**Table 1 vaccines-11-00955-t001:** Summarized Significant Contributions in B-cell M.tb Response.

Topic	Summary	References
**Granuloma**	Phuah et al., 2012 showed the similarities between granulomas and GC in NHP. Their research shows that both structures house B cells in the periphery to create a follicle-like structure. However, unlike GC, only a few B cells within the granuloma were shown to be actively proliferating.	[47,48]
**Acute Infection**	Linge et al., 2022 revealed that knocking out B cell production of IL-6 in mice led to a significantly lower proportion of IFN-γ and Th-17 differentiation. The study was limited, however, due to not providing evidence of IL-6 from B cells after the acute phase of the infection.	[49]
**Chronic Infection**	Linge et al., 2023 used CD-20 monoclonal Abs to KO B cells in mice. Results showed chronic infection of TB in B-cell-depleted mice lead to shorter life spans and increased mRNA expression of neutrophil recruiting factors. These results suggest that B cells may have a protective functionality in chronic stage infection.	[50]
**Rituximab**	Phuah et al., 2016 showed that rituximab-treated NHP had a decrease in IL-6 and IL-10 in their granuloma. However, the results were not statistically significant. Furthermore, a study by Cantini et al. showed that Pt taking rituximab had no increased incidence of TB.	[51,52,53,54,55,56]

Phuah, J.Y., et al., *Activated B Cells in the Granulomas of Nonhuman Primates Infected with Mycobacterium tuberculosis.* American Journal of Pathology, 2012. 181(2): p. 508–514 [47]. Hunter, L.; Hingley-Wilson, S.; Stewart, G.R.; Sharpe, S.A.; Javier Salguero, F. Dynamics of Macrophage, T and B Cell Infiltration Within Pulmonary Granulomas Induced by Mycobacterium tuberculosis in Two Non-Human Primate Models of Aerosol Infection. Front. Immunol. 2022, 12, 776913. https://doi.org/10.3389/fimmu.2021.776913 [48]. Linge, I., et al., *Pleiotropic Effect of IL-6 Produced by B-Lymphocytes During Early Phases of Adaptive Immune Responses Against TB Infection.* Front Immunol., 2022: p. 13:750068 [49]; Linge, I., E. Kondratieva, and A. Apt, *Prolonged B-Lymphocyte-Mediated Immune and Inflammatory Responses to Tuberculosis Infection in the Lungs of TB-Resistant Mice.* Int J Mol Sci., 2023. 24(2) [50]. Brancati, S.; Gozzo, L.; Longo, L.; Vitale, D.C.; Drago, F. Rituximab in Multiple Sclerosis: Are We Ready for Regulatory Approval? *Front. Immunol.* 2021, *12*, 661882. https://doi.org/10.3389/fimmu.2021.661882 [51]; Leandro, M.; Isenberg, D.A. Rituximab—The first twenty years. *Lupus* 2021, *30*, 371–377. https://doi.org/10.1177/0961203320982668 [52]; Monson, N.L.; Cravens, P.; Hussain, R.; Harp, C.T.; Cummings, M.; de Pilar Martin, M.; Ben, L.-H.; Do, J.; Lyons, J.-A.; Lovette-Racke, A.; et al. Rituximab therapy reduces organ-specific T cell responses and ameliorates experimental autoimmune encephalomyelitis. *PLoS ONE* 2011, *6*, e17103. https://doi.org/10.1371/journal.pone.0017103 [53]; Cerny, T.; Borisch, B.; Introna, M.; Johnson, P.; Rose, A.L. Mechanism of action of rituximab. *Anticancer Drugs* 2002, *13*, S3–S10. https://doi.org/10.1097/00001813-200211002-00002 [54]; Phuah, J., et al., *Effects of B Cell Depletion on Early Mycobacterium tuberculosis Infection in Cynomolgus Macaques.* Infect Immun, 2016. 84(5): p. 1301–1311 [55]; Cantini, F., L. Niccoli, and D. Goletti, *Tuberculosis Risk in Patients Treated with Non-Anti-Tumor Necrosis Factor-α (TNF-α) Targeted Biologics and Recently Licensed TNF-α Inhibitors: Data from Clinical Trials and National Registries.* The Journal of Rheumatology, 2014(91:56): p. 56–64 [56].

## Data Availability

Not applicable.

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
