# Peer review of "Role of B Cells in Mycobacterium Tuberculosis Infection"

_vaccines, 2023, doi:10.3390/vaccines11050955_

Round 1

Reviewer 1 Report

Discussion on the role of humoral immunity in TB pathogenesis has a long history. The authors of the article analyze recent publications in the field, concerning such aspects as role of B cells in acute and chronic TB infection, TB granuloma development, vaccination against TB. Despite the fact that the review covers most of the important areas in this field, I believe that it would benefit from a more detailed consideration of the role of memory B cells in the formation of the vaccine effect.

Author Response

REVIEWER#1

Dear Reviewer,

We appreciate you taking the time to review our manuscript.

The manuscript was thoroughly revised by incorporating all your recommendations.

We look forward to publishing this work.

Thanks again for all your feedback.

  1. Discussion on the role of humoral immunity in TB pathogenesis has a long history. The authors of the article analyze recent publications in the field, concerning such aspects as role of B cells in acute and chronic TB infection, TB granuloma development, vaccination against TB. Despite the fact that the review covers most of the important areas in this field, I believe that it would benefit from a more detailed consideration of the role of memory B cells in the formation of the vaccine effect.
  • We agree with your proposal to add more detail about memory B-cells in the vaccine response. As you mentioned, memory B-cells are critical to the immune system's response to vaccines. In response to your suggestions, we added the necessary edits to the paper. We provided more information about the importance of memory B-cells and how they contribute to vaccine efficacy, such as their long lifespan and rapid response to previous exposures. However, to prevent the paper from getting too lengthy, we made the information about memory B-cells concise and short, while not diminishing the importance of memory B-cells. Thank you for your insightful comments and helpful feedback.

Reviewer 2 Report

The authors present an informative update on the status of the role of B cells in TB (both active and latent; acute and chronic).  The review is structured well and logical throughout.  It has the opportunity to be considered for publication.

The following are concerns that can be feasibly addressed.

1. The manuscript is a bit lengthy for the total message that is conveyed, which could be addressed by i) reducing the explanations of basic immunology of APC/T cell/B cell interaction to be more concise and stated once, and ii) reduce redundancies with respect to the statements of reminding the reader that "the role of B cells in TB is not well understood".

2. there is no Fig 2 (only the legend)

3. Lines 289-292:  sentence/statement is confusing, as the KO appears to be of IL-6, so of course IL-6 mRNA would not only be reduced, but should not exist at all (i.e., the name "B-IL6 KO" implies IL-6 was knocked out in addition to a B cell development gene)?  However, if the statement refers only to a mouse in which the "B cell" is KO'ed, then state exactly the KO gene in question.

4. many typos exist, such as the omissions of periods after sentences, strange symbols (such as "&" instead of "and" line 126), periods that do not belong (i.e., line 129), and "AbFc" instead of "Ab Fc" (line 131).

5. Lines 243-244:  check reference 45 - the statement is inaccurate with respect to B cell involvement in T1D and MS in general; i.e., B cell depletion is well known to improve T1D (and MS).  The reference 45 is not supportive of the statement that B cell depletion worsens these autoimmune diseases.

Author Response

REVIEWER#2

Dear Reviewer,

We appreciate you taking the time to review our manuscript.

The manuscript was thoroughly revised by incorporating all your recommendations.

We look forward to publishing this work.

Thanks again for all your feedback.

Reviewer 2:

  1. The manuscript is a bit lengthy for the total message that is conveyed, which could be addressed by i) reducing the explanations of basic immunology of APC/T cell/B cell interaction to be more concise and stated once, and ii) reduce redundancies with respect to the statements of reminding the reader that "the role of B cells in TB is not well understood"

  • We agree that the original draft was too lengthy. We have made significant efforts to address this issue by reducing the word count and ensuring that each section is concise and to the point. However, we believe it is necessary to include basic immunology due to the potential for a non-expert audience to read the journal. While we understand that this may contribute to the overall length of the paper, we believe it is crucial to provide context and ensure that the paper is accessible to a wide range of readers. With that said, we have taken steps to reduce redundancy by using clear and concise language and focusing on the most relevant information while still ensuring that readers have a solid understanding of the topic.

  1. There is no Fig 2 (only the legend)

  • Regarding the lack of a figure in the paper, we want to explain that the figure's absence was due to technical difficulties. Unfortunately, despite our best efforts, we encountered unexpected formatting issues during the transfer process. We understand that the lack of Figure 2 may have impacted the paper's readability, and we apologize for any confusion or inconvenience this may have caused. We assure you that we have taken steps to prevent this from happening again. We have added Figure 2 in the final draft and hope it enhances the reading experience. Moving forward, we will ensure that we have the technical expertise and resources to properly format and transfer all necessary figures and images promptly and efficiently. Thank you again for your feedback.

  1. Lines 289-292: sentence/statement is confusing, as the KO appears to be of IL-6, so of course IL-6 mRNA would not only be reduced, but should not exist at all (i.e., the name "B-IL6 KO" implies IL-6 was knocked out in addition to a B cell development gene)? However, if the statement refers only to a mouse in which the "B cell" is KO'ed, then state exactly the KO gene in question

  • We agree that the original sentence could have been clearer. To better understand the sentence in question. We further explained that experimenters eliminated the development of IL-6 from only B-cells and no other immune cells. To generate B-IL-6KO mice, parental IL-6flox/flox and CD19cre/cre strains were mated, and IL-6flox/-CD19cre/- heterozygous littermates were backcrossed to the parental IL-6flox/flox strain. B-cell development was still maintained in these mice, thus creating B-IL6 KO mice. Since IL-6 is produced by various cells, including macrophages and T-cells, a small amount of IL-6 mRNA will still be reproduced by knocking out only the B-cell IL-6 production. Researchers assessed the total contribution of IL-6 from other immune cells and compared the overall fitness of B-IL6 KO mice to wildtype.

  1. many typos exist, such as the omissions of periods after sentences, strange symbols (such as "&" instead of "and" line 126), periods that do not belong (i.e., line 129), and "AbFc" instead of "Ab Fc" (line 131).

  • We appreciate your attention to detail and your time reviewing our article. Regarding typos and grammatical errors in the paper, we would like to acknowledge that these errors exist. We have eliminated the use of symbols and have used appropriate acronyms. Going forward, we will take greater care in proofreading and editing our work and ensure that we will use appropriate tools to eliminate typos and grammatical errors.

  1. Lines 243-244: check reference 45 - the statement is inaccurate with respect to B cell involvement in T1D and MS in general; i.e., B cell depletion is well known to improve T1D (and MS). The reference 45 is not supportive of the statement that B cell depletion worsens these autoimmune diseases.

  • We would like to address your comment regarding the efficacy of B-cell depletion in improving autoimmune diseases. Upon reflection and further investigation, we must acknowledge that we were erroneous in our original statement that B-cell depletion worsens autoimmune diseases such as T1DBM and MS. You are correct that there is strong evidence to suggest that B-cell depletion can indeed lead to significant improvements in autoimmune diseases such as diabetes and MS. We have made the necessary corrections to fix this mistake. Thank you.

Reviewer 3 Report

In order to better understand the studies as well as the results of the effect of B-cells in the different stages of M.tb infection, the article would be easier to understand with one or two tables in which reference would be made to the results obtained, studies and the bibliographic reference .

Author Response

REVIEWER#3

Dear Reviewer,

We appreciate you taking the time to review our manuscript.

The manuscript was thoroughly revised by incorporating all your recommendations.

We look forward to publishing this work.

Thanks again for all your feedback.

Reviewer 3:

  1. In order to better understand the studies as well as the results of the effect of B-cells in the different stages of M.tb infection, the article would be easier to understand with one or two tables in which reference would be made to the results obtained, studies and the bibliographic reference
  • Thank you for taking the time to review our article and for your valuable comments regarding the inclusion of a table. I appreciate your suggestion, and we agree that a table would be a helpful addition to the article. In response to your comments, we will make the necessary revisions to include a table that summarizes the key findings of the research. Due to technical difficulties, Figure 2 could not be uploaded to the peer review. The absence of the figure greatly impacted the overall readability of the paper. We took the necessary steps to have Figure 2 in the final draft. Including Figure 2 with the new addition of a table will significantly enhance the reading experience. Thank you again for your help.
